# ESAT-6 a Major Virulence Factor of *Mycobacterium tuberculosis*

**DOI:** 10.3390/biom13060968

**Published:** 2023-06-09

**Authors:** Elsa Anes, David Pires, Manoj Mandal, José Miguel Azevedo-Pereira

**Affiliations:** 1Host-Pathogen Interactions Unit, Research Institute for Medicines, iMed.ULisboa, Faculty of Pharmacy, Universidade de Lisboa, Av. Prof. Gama Pinto, 1649-003 Lisboa, Portugal; 2Center for Interdisciplinary Research in Health, Católica Medical School, Universidade Católica Portuguesa, Estrada Octávio Pato, 2635-631 Rio de Mouro, Portugal

**Keywords:** tuberculosis, ESAT-6, ESX-1, virulence factors, PhoPR signal transduction, host-pathogen interactions, TB vaccines, TB diagnosis

## Abstract

*Mycobacterium tuberculosis* (Mtb), the causative agent of human tuberculosis (TB), is one of the most successfully adapted human pathogens. Human-to-human transmission occurs at high rates through aerosols containing bacteria, but the pathogen evolved prior to the establishment of crowded populations. Mtb has developed a particular strategy to ensure persistence in the host until an opportunity for transmission arises. It has refined its lifestyle to obviate the need for virulence factors such as capsules, flagella, pili, or toxins to circumvent mucosal barriers. Instead, the pathogen uses host macrophages, where it establishes intracellular niches for its migration into the lung parenchyma and other tissues and for the induction of long-lived latency in granulomas. Finally, at the end of the infection cycle, Mtb induces necrotic cell death in macrophages to escape to the extracellular milieu and instructs a strong inflammatory response that is required for the progression from latency to disease and transmission. Common to all these events is ESAT-6, one of the major virulence factors secreted by the pathogen. This narrative review highlights the recent advances in understanding the role of ESAT-6 in hijacking macrophage function to establish successful infection and transmission and its use as a target for the development of diagnostic tools and vaccines.

## 1. Introduction

Human tuberculosis (TB) is one of the world’s deadliest infectious diseases [1,2,3]. In 2022, the WHO reported approximately 1.6 million deaths, 10 million new infections, and an estimated one-quarter of the human population being latently infected [3]. Infection control is hampered by the limited efficacy of a 100-year-old Bacille Calmette–Guérin (BCG) vaccine and the emergence of multidrug-resistant strains against 70-year-old antibiotics, particularly those used in first-line therapy [3]. There is also a need to develop better diagnostic tools that are more sensitive, use simple instruments, are easy to handle in remote geographical areas, and are cost-effective for the rapid detection of Mtb. Moreover, although new methods are being implemented [4], the currently available methods do not allow differentiation between the different stages of infection [4,5,6,7,8].

*Mycobacterium tuberculosis* (Mtb) is considered the main causative agent of human TB [9]. Mtb infections primarily affect the lungs, a condition that contributes to high transmissibility by the respiratory route [9]. Although not transmitted from person-to-person, extrapulmonary TB is a second form of disease manifestation that particularly affects less immunocompetent individuals, namely, children or people who are co-infected with HIV [10,11,12]. In addition to Mtb, other pathogens can also infect humans and cause TB with similar clinical symptoms [13,14,15,16,17,18,19,20]. These include *M. africanum*, which is restricted to humans in West Africa [21,22,23], where it causes nearly half of all pulmonary TB [18,19,20], and the animal-adapted species *M. bovis,* which is estimated to cause TB in about 2% of the world’s population [24,25]. *M.caprae* is responsible for a smaller burden of zoonotic TB [26]. They all belong to the *Mycobacterium tuberculosis* species complex (MTBC), which also includes the more distant *M. canettii* group and additional species that cause TB in animals such as *M. microti* [27], *M. pinnipedii*, *M. orygis*, and *M. mungi* [20,23,28,29,30]. Although each of these animal MTBC variants causes TB in its host species, they may trigger slight or no disease outside of their adapted host, especially in immunocompetent hosts [13,30,31]. However, conditions such as the geographic prevalence of infected animals, close human contact, the route of transmission (milk, infected meat, or airborne droplets containing pathogens), and conditions such as HIV infection or other immunosuppressive conditions should be taken into account to assess the true threat to public health [32].

The animal-adapted species are more recent pathogens that emerged as crowd-associated diseases during the Neolithic demographic expansion along with the development of animal domestication [33]. Mtb ascended long before the establishment of crowded populations 70 thousand years ago and accompanied the exodus of Homo sapiens from Africa during the Neolithic expansion [34]. The initial virulence of Mtb is therefore strongly adapted to the occasional availability of the population to be infected and to an evolutionary ability to survive for a long time in the host until the opportunity for transmission arises. Consequently, Mtb infections have evolved in low human population densities, exhibiting a pattern of chronic development, accompanied by decades of latency before progressing to active disease [10,11,12]. It is a well-adapted human pathogen, requiring the induction of a strong inflammation and destruction of lung tissue for transmission and evolutionary survival [9,34,35]. This feature is unusual in most pathogens, where virulence is not associated with their spread to other hosts [36].

The pathogen does not seem to require virulence factors such as the usual pili, toxins, or capsules that are essential for the invasion of epithelial barriers [9]. Nevertheless, their potential involvement should not be ignored, as a capsule and pili have recently been identified in Mtb, although their roles during infection in vivo are still unclear [37,38,39]. Mtb virulence relies on its ability to manipulate host macrophages, where it establishes intracellular niches to cross mucosal barriers and avoid pathogen destruction. First, Mtb subverts the endocytic pathway, preventing phagolysosome fusion and proteolytic digestion [40,41,42,43,44,45]. Second, it activates innate immune responses to induce its transmigration into the lung parenchyma [46,47]. There, infected macrophages attract more permissive cells, expanding intracellular niches [9,48,49,50]. Mtb induces the adaptive responses that stimulate its containment and encourage a long life inside granulomas [51,52,53,54]. Finally, the pathogen induces necrotic cell death in macrophages, granuloma destruction, and lung cavitation for transmission [47,49,55,56]. Common to all these events is the major virulence factor: the “early secreted antigenic target of 6 kDa” (ESAT-6, also called EsxA). This review highlights the role of ESAT-6 in different phases of Mtb infection and its contribution to virulence. It also points to ESAT-6 as a target for the development of better diagnostic tools and future vaccines for human tuberculosis.

## 2. Virulence Evolution among MTBC

A comparative genomic analysis among MTBC species reveals more than 99.95% sequence homology [57], but they differ mainly by large sequence polymorphisms [18] relative to Mtb, reflected by the so-called regions of difference (RD) and translated into deletions [57,58,59,60,61] or punctual insertional sequences [62]. These observations reinforce the ancestral origin of Mtb, reflecting a loss of genes during the transmission to animals that enabled a fitness gain in the new host and a loss of robustness in humans. In the case of *M. africanum*, lineage 5 (L5) is more associated with Mtb-like lineages, while lineages L6 and 9 display an RD that is more associated with *M. bovis*-like animal-adapted species [19,20].

Important cumulative findings from these studies have identified more pathogenic MTBC disease-causing species from less fit ones with deletions/insertions/mutations in the RD regions affecting the PhoPR two-component virulence system or the genes under the control of this signal transduction regulator [63,64]. This includes the regions called the region of difference 1 (RD1) and the region of difference 8 (RD8), which are responsible for the production and secretion of the virulence factor ESAT-6 [29,62].

Particularly, the RD1 region is deleted in the animal-adapted species *M. microtii* and was first described in BCG, where it is absent and associated with the attenuation of this live vaccine during its propagation in vitro [23,65,66]. The RD8 deletion is associated with animal-adapted species, and it affects the regulatory PhoPR-dependent region of the *espACD* operon involved in ESAT-6 secretion (Figure 1). The genetic transfer of mutations affecting the PhoP binding region in *M. bovis* and in the closely related *M. africanum* L6 into the Mtb *sensu stricto* human species resulted in reduced ESAT-6 secretion and lower virulence [29]. Remarkably, the deleterious effects of these mutations were partially compensated by RD8 deletions in both species, allowing ESAT-6 secretion to some extent by creating alternative regulatory sequences [62]. The observed attenuated ESAT-6 responses contribute to the observed slower clinical progression from infection to disease when compared to Mtb [18,67]. Conversely, the insertion of the IS*6110* element upstream of the PhoP binding locus resulted in the upregulation of the operon in one multidrug-resistant *M. bovis* strain, responsible for an unusually high human transmissibility and partially reversing the *phoPR-bovis*-associated fitness loss [68,69].

The ESAT-6 gene (*esxA*) is part of the *esx-1* locus, a group of genes encoding the type VII secretion system that allows the secretion of the virulence factor ESAT-6 from the pathogen, known as the ESAT-6 secretion system 1 (ESX-1) [70,71] (Figure 1).

## 3. ESAT-6 Is Required for the Virulence of Mtb

ESAT-6 was first identified in 1995 by the discovery of a potent T-cell antigen in the short-term culture filtrate of Mtb [72,73]. The initial focus was on its potential use as a target for a TB vaccine to replace BCG [74]. Moreover, since patients and animals infected with MTBC respond strongly to ESAT-6 antigens, parallel emphasis was given to the development of a better diagnostic tool for TB [75].

The export of ESAT-6 from the bacilli requires a secretory apparatus consisting of proteins that are assembled on the inner surface of the cell membrane and are strongly recognized by T cells [70,71,76]. The encoded genes are all part of the *esx-1* operon (Figure 1), and there is increasing evidence that they are under selective pressure imposed by the host’s immune system [76]. Concomitant with these studies, comparative functional genomics among virulent, attenuated, or saprophytic mycobacterial species have contributed to the inclusion of ESAT-6 in Mtb virulence [77]. Curiously, the saprophyte species *M. smegmatis* (Ms) also encodes for an ESX-1 apparatus [78]; however, it does not appear to confer Ms virulence capabilities, as demonstrated by its inability to survive in human macrophages [79] or in amoeba in the environment [9]. The emergence of ESX-1-associated virulence in Mtb stems from the original role of ESX-1 in Ms and other GC-rich soil species in inducing horizontal gene transfer between bacteria [80,81,82]. Predatory amoeba may have contributed to the evolutionary pressure that selected mycobacterial pathogens for intracellular survival. ESAT-6 requires an extended *esx-1* locus to be secreted, a putative genomic island containing the *espACD* locus, which is adjacent to the region of difference 8 (RD8) (Figure 1). It has been hypothesized that this phenotype may be associated with independent horizontal gene transfer from pre-pathogenic mycobacteria in contact with soil bacteria and positively selected while in amoeba phagosomes [83,84].

ESAT-6 secretion from the bacilli requires both the expression of the *esx-1* locus for the type VII secretion apparatus and the transcription of both the ESAT-*6* gene (*esxA*) and the culture filtrate protein 10 (CFP-10) gene (*esx-B*) contained in the RD1 region. In addition, it requires the protein EspA, which is not encoded in the *esx-1 locus* but in the extended *espACD* operon adjacent to RD8 [85] (Figure 1), which is located 260 kb upstream of *esx-1* [71].

All species and strains deleted in the *esx-1* locus, the internal RD1 region, or the *esx-1* extended locus *espACD* exhibit an attenuated phenotype [86]. Mutants with deletions on ESX-1 of Mtb are attenuated in virulence, translating into reduced survival of mycobacteria in cultured macrophages or in experimental animal models of TB [87,88,89,90,91,92], which is consistent with the attenuation of the BCG vaccine or the species *M. microti* due to the deletion of the region of difference RD1 [65,66]. Likewise, the introduction of RD1 into BCG restores virulence [90,93]. The reference laboratory strain Mtb H37Ra has a mutation in the *phoP* regulatory region that is responsible for the attenuated phenotype compared to its virulent counterpart Mtb H37Rv, resulting in impaired ESAT-6 secretion [63,94]. Clinical Mtb isolates were shown to secrete higher levels of ESAT-6 than the reference laboratory strain Mtb H37Rv. A comparative analysis of genetic polymorphisms between clinical and laboratory strains revealed *whiB6* (*rv3862c*), a gene upstream of the ESX-1 genetic locus. It encodes for a regulatory protein that activates promoters under the *esx-1* and extended *espACD* locus responsible for ESAT-6 production and secretion, respectively [95] (Figure 1).

Clearly, the loss or gain of mycobacterial virulence is closely linked to the ability of mycobacteria to produce and secrete ESAT-6, and the extension of virulence is correlated with the amount of protein secreted.

## 4. ESAT-6 in Mtb Pathogenesis

### 4.1. During the Early Phases of Infection: The Innate Phase

Upon inhalation of aerosols containing the pathogen, the bacilli that manage to reach the alveoli are phagocytosed by alveolar macrophages [10]. These professional phagocytic cells are permissive to Mtb and provide a critical niche for the survival of the intracellular pathogen [9,44,96,97]. In the endocytic pathway, the pathogen inhibits phagosome fusion with lysosomes and prevents vesicle acidification and proteolytic destruction using lysosomal enzymes that allow the pathogen to replicate in early phagosomes [40,41,42,43,44,45]. Recent studies have shown that ESAT-6 inhibits IL-18-mediated phagolysosome fusion by regulating microRNA-30a in mycobacteria-infected macrophages [98,99]. ESAT-6 was indeed associated with the blockade of phagosomal maturation because mutants in ESAT-6 of *M. marinum*, a pathogen that causes a TB-like disease in fish, were found to be mainly located in lysosomes in contrast to the wild-type strain, which is located in early phagosomes [100].

However, luminal studies have challenged the dogma of the exclusive intracellular localization on phagosomes of both *Mtb* and *M. marinum* [101,102,103,104,105]. Studies using ultrastructural observations generated by electron microscopy and, more recently, the development of a Fluorescent Resonance Energy Transfer (FRET) method demonstrate the translocation and escape of both pathogens from phagolysosomal compartments into the cytosol [101,102].

It has been shown that ESAT-6 induces phagosomal membrane rupture, allowing pathogens to gain access to the cytosol, in contrast to the corresponding mutants, which instead accumulate in phagosomes. Moreover, following phagolysosomal escape, a necrotic form of cell death in infected macrophages was observed 3 to 4 days post-infection in ex vivo studies [101].

It is unlikely that in all contexts of infection in vivo, ESAT-6 will induce complete rupture of the phagosome membrane and total escape of the bacteria into the cytosol, ending with necrotic cell death. Necrotic cell death occurring in vivo will induce a strong inflammatory response and tissue destruction. There is a possibility that punctual membrane perturbations may create local conditions for the transfer of Mtb proteins with less induced stress in the cytosol. ESAT-6 is a pathogen-associated molecular pattern (PAMP) sensed by innate cytosolic receptors and is the major Mtb PAMP that activates the NLRP3 inflammasome [47]. This platform is required for caspase-1-mediated processing of the cytokines IL-1β and IL-18. The combination of purified ESAT-6 protein with other PAMPs such as Ag85 has a significant impact on NLRP3 activation and IL-1β secretion, demonstrating that ESAT-6 helps other Mtb PAMPs reach the cytosol [47]. IL-1β drives neutrophil recruitment by several mechanisms, and if not controlled, neutrophils are major instructors for tissue destruction [55,106]. The mechanism is controlled by nitric oxide (NO) released by IFN𝛾-activated macrophages [55,106]. NO in turn controls the inhibition of the NLRP3 inflammasome, which regulates the amounts of cytokines secreted.

Independent studies have shown that NLRP3 activation does not always result in the necrotic cell death in infected cells and tissue destruction. The transmigration of infected alveolar macrophages (AMs) into the lung parenchyma is dependent on Mtb ESX-1 inducing IL-1β via the NLRP3 inflammasome. As a result, IL-1R signaling on alveolar pneumocytes affects alveolar permeability and lung tissue access without causing tissue destruction [46,47].

Once inside the lung, infected macrophages activate pneumocytes surrounding the nascent granuloma to secrete matrix metalloproteinase 9 (MMP9) in an ESAT-6-dependent manner [107,108,109]. Consequently, an influx of more permissive macrophages following the MMP9 signals reaches the nascent granuloma and efficiently finds and performs efferocytosis of dying apoptotic infected cells [110,111]. Continuous cycles of this process allow the expansion of Mtb intracellular niches [107,108,112]. The intracellular replication and bacterial load are controlled by distinct mechanisms, some of which depend on perturbations of the ESAT-6 phagosomal membrane with concomitant cytosolic PAMPs that activate different antimicrobial mechanisms, including apoptosis [48,110] and autophagy [113]. The ability to induce apoptosis is a feature of the virulent strains of *M. tuberculosis* in a process that involves ESAT-6 [50,114]. Moreover, it has also been shown that the inhibition of apoptosis by non-virulent mutants of *M. marinum* impairs the spread of infection and bacterial expansion in nascent granulomas [52]. The apoptotic form of controlling intracellular bacterial loads counteracts the necrotic forms of death that are usually observed during high-load infections of macrophages in vitro [115] and contribute to the cell-to-cell spread of Mtb.

Autophagy is a relevant innate response that controls intracellular pathogens, including Mtb [116]. ESAT-6 contributes to this pathway [113,117,118]. Its functions in perturbing the phagosomal membrane will expose PAMPs, including mycobacterial DNA, a signature that can be sensed by host innate cytosolic receptors. These include at least three cytosolic sensors, two involving the inflammasomes NOD-, LRR-, and pyrin domain-containing 3 (NLRP3) and absent in melanoma 2 (AIM2), and the third the cyclic GMP-AMP synthase (cGAS) [117]. The latter leads to the synthesis of the second messenger cyclic GMP-AMP (cGAMP), which activates the endoplasmic reticulum-associated stimulator of interferon genes (STING) and the downstream serine/threonine-protein kinase (TBK1)-interferon regulatory factor 3 (IRF3)–IFN-I signaling pathway (Figure 2) [113,119]. TBK1 provides a bridge for Mtb destruction by targeting intracellular bacteria to the ubiquitin-mediated autophagic pathway in macrophages [113].

### 4.2. During Latency

The latency period can last for decades after the initial infection. It is defined as a state of immune activation in the absence of disease symptoms [11]. During the innate granuloma expansion, the adaptive responses are activated, and the latency phase begins with the arrival of effector T-lymphocytes to the granuloma [10]. The adaptive granuloma is a structure that contains Mtb in the lung, but it is also the result of a strategic manipulation by the pathogen to ensure a long life in the host. A fine balance of proinflammatory microbicidal mechanisms with immunosuppressive events drives the process. Bacterial loads are kept at bay in macrophages by effector CD4^+^ T cells, such as Th1 secreting the cytokines IFN𝛾 and TNFα, Th17 secreting IL-17, and in much smaller numbers, Th2 and regulatory T cells that counteract the inflammatory effects of the former [10]. The adaptive T cells are constantly arriving at the granuloma forming a surface layer and a few infiltrating cells will license-infected macrophages to exert their activity. In the case of Th1, this licensing induces macrophages to secrete IL-12 allowing efficient IFN𝛾 and TNFα release from effector T cells. Mtb ESAT6 has been shown to induce immunosuppressive regulatory T-cell populations that delay effector T-cells migration into the granuloma, a process that is reversed by IL-12 [120]. In turn, IFN𝛾 and TNFα activate infected macrophages to become more microbicidal. However, Mtb resists these effects by reducing macrophage responsiveness to signaling by IFN𝛾 [121]. IFN𝛾 also contributes to NO release from activated macrophages, a mechanism required to control IL-1β and tissue destruction, helping to preserve the structure of the granuloma [55].

In the case of CD8^+^ T-lymphocytes, ESAT-6 in infected macrophages interacts with beta-2-microglobulin (β2M) in the endoplasmic reticulum, affecting antigen presentation to effector cytotoxic T cells and impairing their microbicidal activity [122,123].

While macrophages become more microbicidal, the pathogen activates the two-component PhoPR system, which allows the pathogen to adapt to a stressful environment, such as hypoxia and low pH, while activating the dormancy regulon that puts the bacilli in a low metabolic state [124].

Activated macrophages secrete TNFα and IL-1β, which permeabilize the endothelia to feed the granuloma with newly arrived macrophages and neutrophils to be infected. All of this cell arrival dynamic is fueled by new blood vessel formation induced by an ESAT-6-driven release of the angiogenesis factor VEGF from the infected cells in the granuloma [125].

### 4.3. During Progression to Disease

Although mycobacteria exploit macrophages to maintain a long life in intracellular niches within a host, they promote their transmission to a new host by becoming extracellular [9]. At this stage, the promotion of necrotic cell death in highly infected cells allows the release of the pathogen by a process requiring ESAT-6 [115]. Moreover, as stated before, apoptosis induced by virulent Mtb favors the intracellular expansion of the pathogen but with a reduced load per host cell [48,110], whereas intracellular Mtb replication tends to enhance necrosis [49,126]. Moreover, it has been demonstrated that Mtb is able to survive in this necrotic environment. As mentioned above, ESAT-6-mediated access of the pathogen PAMPs to the cytosol, including DNA, activates several cytosolic sensors including the NLRP3, the AIM2, and the cGAS. The first two are involved in the secretion of IL-1β, while the latter leads to the release of type I interferon (IFN-I). The crosstalk between IFN-I and IL-1β influences the progression of the disease by controlling cell death within the granuloma [117,127,128].

Observed high levels of IFN-I in the serum of infected patients have been associated with progression to TB [129]. Type I IFNs subvert anti-tuberculous host defenses by inhibiting iNOS, the enzyme responsible for NO production, while inducing the immunosuppressive mediator IL-10 [127]. Based on the stimulation of iNOS by IFNγ and the inhibition of iNOS by IFN-I, it appears that an imbalance of IL-1β results from NLRP3 inflammasome activation [55]. Under IFNγ control, IL-1β enhances TNFα-stimulated Mtb killing and contributes to controlled neutrophil recruitment. In the context of low levels of TNFα, IL-1β stimulates cyclooxygenase-2 (COX2) to produce prostaglandin E2 (PGE2) from arachidonic acid (AA), resulting in mitochondrial membrane protection and controlled intracellular burdens via apoptosis [130]. Unlike IL-1β, IFN-I stimulates 5-lipoxygenase (5-LO), which is a competitive enzyme for COX2, to produce lipoxin A4 and leukotriene B4 from AA, which leads to the loss of plasma membrane integrity, cytoplasmic organelles swelling, such as mitochondria and nuclei, and making cells more susceptible to necrotic cell death.

Unexpectedly, the excessive TNFα in the granuloma that is often observed during active TB promotes a necrotic form of death called necroptosis via the production of mitochondrial reactive oxygen species (ROS) [127,128,131].

The bacteria released into the caseous center of the granuloma find a nutrient-rich environment for massive replication. This high lipid content is mostly the result of the necrotic cell death in infected foamy macrophages in an ESAT-6-dependent manner [132].

Overall, IL-1β, in synergy with TNFα, is a major inducer of neutrophil recruitment to the lung and to the granuloma. In active TB patients, neutrophils represent the major infected cell population [49]. A synergistic mechanism can be attributed to the fact that IL-1β is required for Th17 polarization, and the cells releasing IL-17 will activate endothelia to release chemokines for neutrophil recruitment. However, transmigration occurs only after endothelial E-selectins are exposed following stimulation by both cytokines [128,133]. Neutrophils are one of the main causes of pathological tissue sequelae due to bioactive neutrophil molecules, including proteases and metalloproteinases [53,134]. Indeed, Mtb in infected cells induces human neutrophils necrosis in an ESAT-6-dependent manner, and neutrophil-produced reactive oxygen species (ROS) drive this necrosis [49]. Impaired dead cell clearance leads to severe tissue inflammation and contributes to the granuloma disruption and lung cavitation required for the subsequent transmission of infection to the next host [135]. The involvement of ESAT-6 in mediating the inflammation required for transmission is indeed supported by studies showing its increased expression in the highly transmissible Mtb Beijing lineage [136], as well as in a mutant of the *M. bovis* strain, which is responsible for a human outbreak in 1992 [68,69].

## 5. ESAT-6 from a Virulence Factor to Diagnostic Tools and Vaccines for TB

Research on ESAT-6 and its involvement in several steps of Mtb pathogenesis, together with the strong antigenic recognition in TB patients, reveals its potential for therapeutic and diagnostic applications.

ESAT-6, through its duality of virulence and antigenicity, is a target for the design of more effective vaccines than BCG. The strategic design of new live attenuated vaccines should preserve Mtb antigens while removing virulence factors to prevent host damage. MTBVAC, a vaccine in development that has just entered phase 3 clinical trials (see [137]), is the only vaccine based on an attenuated Mtb strain [138]. It is conceivable that MTBVAC, by targeting epitopes from RD1 that are missing from the BCG vaccine, could provide better protection against TB.

The attenuated virulence phenotype is based on PhoP mutants that are unable to secrete ESAT-6. The protein is synthesized but it remains inside the pathogen with its antigenic potential. PhoP mutations in the attenuated vaccine strain prevent EspA translation from the *espACD* locus. A second deletion affects the gene required for the biosynthesis and export of phthiocerol dimycocerosates (PDIM), the major virulence-associated cell-wall lipids of Mtb. Both EspA and PDIM act together in the phagosomal secretion of ESAT-6 [105,139].

Subunit vaccine candidates are designed to boost BCG-primed responses or induce specific immune responses as examples of the therapeutic vaccines under development to be used in conjugation with antibiotic treatment [140,141,142,143]. ESAT-6-based subunit vaccines that are already in clinical trials include TB/FLU-04L and use a live-attenuated influenza A virus vector. Other subunit vaccines in clinical evaluation, such as H6, H56:IC31, and GamTBvac are provided in non-viral delivery systems and are based on fusion immunogenic proteins, including ESAT-6 combined with adjuvants [142,144,145,146].

An intranasally administered subunit vaccine combining ESAT-6 and cyclic dimeric adenosine monophosphate (c-di-AMP) was designed to promote macrophage autophagy via the STING pathway with an impact on pathogen killing and humoral and cellular immune responses [147]. Other fusion proteins, such as dodicin-ESAT-6, result in increased expression of the costimulatory molecules CD80/CD86 and the antigen-presenting machinery MHC-II in a mouse model of the infection [148]. In the improvement of these combinations with ESAT-6 in subunit vaccine candidates, the chaperone-like protein HtpG_Mtb_ of Mtb,= has been studied, and structure-based design studies point to more effective antigen properties for immunization [149,150,151].

Some challenges in the development of ESAT-6-based subunit vaccines are that while CD4 T cells are maintained in the lung parenchyma due to continuous antigenic stimulation, the protective immunity is limited by functional exhaustion [152].

Another relevant application of ESAT-6 has been the development of TB diagnostics such as the IFNγ release assays (IGRAs) [153]. They were initially designed to guide preventive treatment of infected individuals at risk of developing active TB. Additionally, the ESAT-6-based IGRA allows differentiation between BCG-vaccinated and unvaccinated individuals, as BCG does not possess or secrete these proteins.

However, the challenges associated with subunit ESAT-6-based subunit vaccines are that they will virtually override all modern available immunodiagnostics (IGRAs and skin tests), thereby limiting the ability to distinguish immunized from infected people. To overcome this, ESAT-6-free IGRA tests are under development, aimed at differentiating ESAT-6 subunit responses in vaccinated individuals. These assays are based on the combination of a cocktail of proteins that are part of the ESX-1 operon, including the CFP-10 chaperon but excluding ESAT-6. Preliminary results show promising effects on the induction IFNγ and the chemokine biomarker IP-10, distinguishing subunit-vaccinated individuals from non-vaccinated individuals [75].

An ESAT-6/CFP-10-based skin test, C-Tb, has revealed a similar sensitivity for active TB compared with tuberculin skin test (TST) and QuantiFERON-TB-Gold-In-Tube (QFT), but with limited sensitivity in children and HIV-infected individuals [154]. The diagnosis of TB in children as in HIV-infected individuals is often difficult due to several factors, including the frequent extrapulmonary TB, a recurrent associated sputum smear-negative, and the fact that they have a low humoral response to mycobacterial antigens using conventional enzyme-linked immunosorbent assays (ELISA) for TB immunodiagnosis [51,155]. To overcome this, immunodiagnostic methods using antigen combinations are being developed, using “cocktails” of PhoP, ESAT-6, CFP-10, and the latency-associated antigen Acr-1. This combination of antigens includes proteins that are associated with the different stages of disease progression [156] and has been shown to significantly improve sensitivity. Recently, another cocktail using biosynthetically derived peptides of ESAT-6 and Ag-85 allowed the detection of irrespectively specific IgG in patients’ sera, providing a reliable diagnosis of active TB in children [157].

Other situations where extrapulmonary TB is frequent are bovine TB and the corresponding zoonotic form in humans. The available methods to distinguish *M. bovis* from *Mtb* are based on polymerase chain reaction and genomic sequencing. This imposes limitations requiring bacilli isolation, which is difficult to obtain in this context, and a lack of technical capacity in high-burden countries [32].

Future directions on diagnostic tools should allow us to determine differences between latent TB infection, BCG/sub-unit vaccinations, active TB infections, extrapulmonary TB with enough sensitivity to detect all cases of TB, including children and HIV-infected people, and MTBC species. This could revolutionize TB diagnostics and treatment strategies allowing the development of differentiating biomarkers that are relevant for evaluating the status of immune activation and/or the stage of the infection.

## 6. Conclusions

Studies on ESAT-6 have allowed us to define the relevant steps in Mtb pathogenesis and to distinguish virulence from attenuated phenotypes. The lessons from this knowledge will allow us to foster our understanding of this proteinaceous army that makes Mtb such a successful human pathogen. The likelihood that this could enable us to target ESAT-6 and the host hijacking pathways involved to halt the spread of disease is a possibility for the near future. Perhaps this will open new avenues leading to the development of novel immunotherapeutic strategies to stop TB in the 21st century.

## Figures and Tables

**Figure 1 biomolecules-13-00968-f001:**
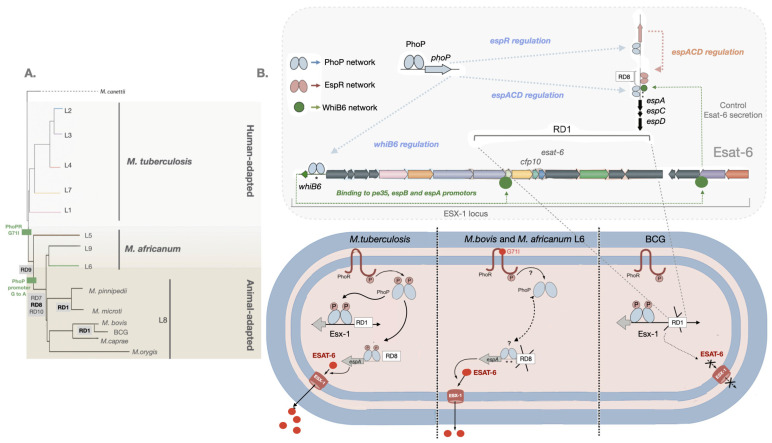
Control of ESAT-6 secretion by PhoP-dependent regulatory networks. (**A**) Lineages of the MTBC with RD deletions and PhoPR mutations are highlighted [20,23,28,29]. (**B**) Several genes from the ESX-1 and the extended ESX-1 (*espACD* and *espR*) regions required for ESAT-6 protein synthesis and exports are displayed. PhoP (blue ellipses) is a transcriptional activator and an effector of the signal transducer PhoR that interacts with the *espR*, *espA*, and *whiB6* promoters. In its phosphorylated form, PhoP binds to DNA with higher affinity. Mtb, carrying a functional PhoR, is able to sense a stress-like stimulus and subsequently phosphorylate PhoP. EspR (pink ellipses) activates the *espACD* locus. WhiB6 (green circles) also interacts with the promoter regions of the *espA*, *pe35*, and *espB* genes. The *espACD* locus is activated by all these circuits, and the protein translated EspA is required for ESAT-6 secretion. The RD1, absent in BCG, and RD8, absent in *M. africanum* L6 and L8 animal-adapted lineages (including *M. bovis*), as well as polymorphisms in the *espACD* and *whiB6* promoters (asterisks), are indicated. *M. bovis* and *M. africanum* L6, carrying a defective PhoR G71I allele, are expected to generate a low-affinity binding effector due to phosphorylation impairment of PhoP. Nevertheless, ESAT-6 secretion in these species is restored to some extent by compensatory mutations in the *espACD* promoter region, including RD8 deletions and species-specific polymorphisms (asterisks) [62].

**Figure 2 biomolecules-13-00968-f002:**
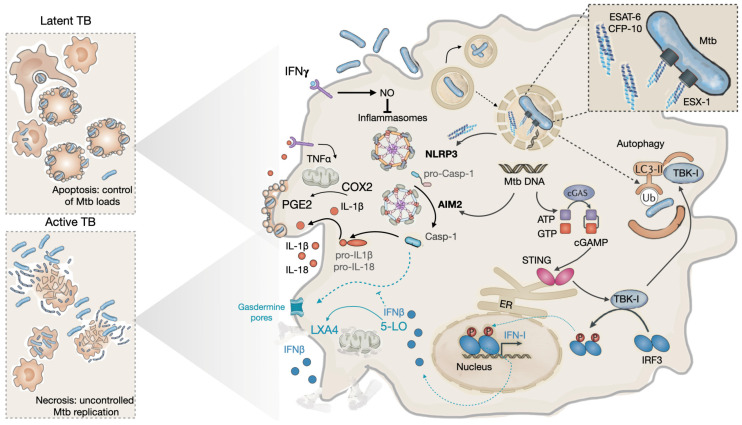
Effects of ESAT-6 on the outcome of Mtb infection in host macrophages. Upon internalization into macrophages, Mtb manipulates the endocytic pathway to prevent fusion of pathogen-containing phagosomes with lysosomes. This allows the bacilli to replicate in early phagosomes (top of the figure). A few pathogens escape into the cytosol via the ESX-1 secretion apparatus, which secretes ESAT-6. ESAT-6 induces perturbations in the phagosomal membrane, allowing bacilli or their PAMPs to exit the vesicle. PAMPs such as ESAT-6 or bacterial DNA activate innate sensors such as NLRP3 and AIM2 to form inflammasomes required for IL-1β processing via casp-1. Under IFN𝛾 signaling regulated amounts of cytokines are secreted from of the cell, allowing the control of the intracellular bacterial loads by TNFα, which in turn induces apoptosis. In addition, IL-1β activates COX2 to produce PGE2 an eicosanoid that protects the mitochondrial inner membrane and contributes to apoptosis. Alternatively, bacterial DNA activates the cGAS/cGAMP/STING pathway, leading to TNK-1 activation, which will induce autophagy and therefore control the intracellular load of Mtb. This is what mostly happens during latent TB (black arrows). Under low IL-1β, controlled recruitment of neutrophils prevents tissue damage. During progression to active TB disease, bacterial replication induces high levels of TNFα, and this, together with impairment of IFN𝛾, results in the strongest activation of inflammasomes; casp-1 stimulates gasdermin to form pores at the cell membrane inducing pyroptosis. High levels of TNFα induce necroptosis. TBK-1 phosphorylates the transcription factor IRF3 leading to the secretion of type-I IFN such as IFNβ. IFNβ is immunosuppressive in chronic infections by activating IL-10, deactivating macrophages, and promoting LX4 synthesis via 5-LO. LX4 causes organelles such as mitochondria to swell and induces cell death by necrosis. Altogether, this helps Mtb replicate massively in macrophages and release it into the extracellular environment (light blue lines).

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
