# Peer review of "ESAT-6 a Major Virulence Factor of Mycobacterium tuberculosis"

_biomolecules, 2023, doi:10.3390/biom13060968_

Round 1

Reviewer 1 Report

The manuscript refers to one of the most important human diseases, but the content level is not sufficient.

The authors do not mention the review type, nor the criteria used for the selection of the presented information, please modify accordingly.

Several statements are not entirely accurate, authors are recommended to upgrade the scientific level:

1.      Line 33 “diagnostic based, in vast part of the world, on a 120-year-old microscopy technique [4,5].“  the protocols to confirm and monitor TB patients are complex and yes, collaborated with other results the microscopic evaluation is still a valid method.

2.      Lines 35-44 require drastic rephrasing: there are too many word repetitions and a mixture of information – etiology, epidemiology, and history, while the epidemiology vocabulary is not used with accuracy.

The authors should list the etiological agents of human tuberculosis – please, add also Mycobacterium avium and state that there are particularities of occurrence, severity of clinical signs, outcomes, geographical location, occupational hazard etc in relation to the associated etiological agents.

“animal-adapted species such as M. bovis”? scientific literature describes M. bovis not only in cattle, but also other domestic and wild species

3.      Lines 51-55 “accompanied by decades of latency before progressing to active disease [17-19]. It is a well-adapted human pathogen, requiring the induction of a strong inflammation and destruction of the lung tissue for transmission and evolutionary survival [6,16]. This feature is unusual in most pathogens where virulence is not associated with their spread to other hosts [20]. “ should be modified.

The phrases suggest that the tissue modification induced by MTBC are characteristic for MTBC, but the cell mediated response in described in case of Mycobacterium bovis and Mycobacterium avium. The lung is not the only organ that presents the modifications.

4.      What is the connection between virulence factors and transmission to other hosts?

MTBC presence and consequent tissue modifications are described in several other species and the MTBC is known as a reverse zoonosis for both domestic and wild animals.

5.      No link between the lines 43-55 and the aim of the study.

6.      The authors do not underline the novelty of the study.

7.      Lines 71-100 present a mixture of information

The authors should refer to ESAT-6 as a virulence factor in MTBC and present the differences found when comparing the genome of other species considered as TB etiological agents.

8.      Lines 124-350 should follow the introduction, while lines 71-100 should continue these

9.      lines 352-385 should be developed 

the English language level is sufficient

Author Response

The manuscript refers to one of the most important human diseases, but the content level is not sufficient. 

The authors do not mention the review type, nor the criteria used for the selection of the presented information, please modify accordingly. 

This manuscript focuses on human tuberculosis, not on TB in other animals. The species Mtb referred on the title and on the first phase of the abstract or the first sentence of the Introduction clearly contextualize this because M. tuberculosissensu stricto only cause TB in humans. In addition, the word TB and successful human-adapted pathogen means the ability to live long without causing disease in the host plus the ability to generate disease in order to be transmitted to other humans. The concept of of virulence factors in Mtb refers to these two contexts. Likewise, ESAT-6 is a major virulence factor of Mtb because in humans it intervenes in all relevant phases of the infection. The novelty of this review is based on the highlightening of these different phases in a single manuscript with the most recent publications regarding to angiogenesis or induction of necrosis and survival during the caseating and cavitation phase published after 2016. I did not find a single review published that addresses all these data together. The reference to therapeutic applications is not the aim of the manuscript but the consequent application of the generated ESAT-6 knowledge. To avoid misinterpretation, we rename the title to “ESAT-6 a major virulence factor of Mycobacterium tuberculosis”.

Several statements are not entirely accurate, authors are recommended to upgrade the scientific level:

  1. Line 33 “diagnostic based, in vast part of the world, on a 120-year-old microscopy technique [4,5].“  the protocols to confirm and monitor TB patients are complex and yes, collaborated with other results the microscopic evaluation is still a valid method.

R: We agree with the reviewer that it is a valuable diagnostic tool, but it is limited, as stated in the manuscript published in The Lancet by Dedra 2016 (reference 4): “Despite suboptimum sensitivity (∼50%; limit of detection of ∼104 organisms per mL), smear microscopy is the standard of care in most high-burden settings”. However, we agree that the word “vast majority” is excessive. We now rephrase for:” There exists also a need for better diagnostic tools to discriminate between different infection statuses that can reach all geographical areas. In high burden geographic areas such as some regions of Africa, diagnosis is based on a low-sensitive, 120-year-old microscopy technique [4,5]”.

  1. Lines 35-44 require drastic rephrasing: there are too many word repetitions and a mixture of information – etiology, epidemiology, and history, while the epidemiology vocabulary is not used with accuracy.

The authors should list the etiological agents of human tuberculosis – please, add also Mycobacterium avium and state that there are particularities of occurrence, severity of clinical signs, outcomes, geographical location, occupational hazard etc in relation to the associated etiological agents. 

“animal-adapted species such as M. bovis”? scientific literature describes M. bovis not only in cattle, but also other domestic and wild species 

  1. We agree with the reviewer that there is a mix of information regarding epidemiology and etiology, in those statements that we rephrase to address the concerns:

“Other pathogens can also infect and cause TB in humans with lower levels of disease morbidity and are rarely transmitted from person-to-person. These, include M. africanum and animal-adapted species such as M. bovis, M. microtii or M.caprae. The former is restricted to humans in West Africa [7], where it causes nearly half of all pulmonary TB cases [8-10], while M. bovis is estimated to cause TB in about 2% of the world’s population [11], and both M. microti and M.caprae rarely infect humans.  They all belong to the Mycobacterium tuberculosis species complex (MTBC) which also includes the more distant M. canettii group and species that cause TB only in animals, such as M. pinnipedii and M. orygis [10,12-14].”

Regarding M. avium, it is not a member of MTBC and is a non-tuberculous bacteria (NTB): as stated in the link by Edward A. Nardell, MD, Harvard Medical School actualized in 2022:

 “Tuberculosis is caused by bacteria called Mycobacterium tuberculosis. Other related bacteria (called mycobacteria), such as Mycobacterium bovis or Mycobacterium africanum, occasionally cause a similar disease. These bacteria plus Mycobacterium tuberculosis and some others are called the Mycobacterium tuberculosis complex. Other mycobacteria, particularly the group called M. avium complex (MAC), also cause disease in people. The diseases they cause are different from tuberculosis.(https://www.msdmanuals.com/home/infections/tuberculosis-and-related-infections/tuberculosis-tb) (https://www.msdmanuals.com/home/infections/tuberculosis-and-related-infections/infections-caused-by-bacteria-related-to-tuberculosis-tb.

In fact, M. avium does not cause granulomas in humans and moreover does not express ESX-1, including ESAT-6 (https://doi.org/10.1128/CDLI.6.4.606-609.1999; http://dx.doi.org/10.1016/j.cell.2014.11.024 ).

Concerning M.bovis as the referee pointed out, it is as an animal-adapted species; we did not mention the restriction to cattle, but cattle are the major responsible for transmission to humans unlikely all other wild species, because the close contact with a lot of animals of large dimension and the historically reported milk transmission before pasteurization or uncooked meet. And we contextualized the historical part here with all accuracy: “The animal-adapted species are more recent pathogens that emerged as a crowd-associated diseases during the Neolithic demographic expansion along with the development of animal domestication [15].”

  1. Lines 51-55 “accompanied by decades of latency before progressing to active disease [17-19]. It is a well-adapted human pathogen, requiring the induction of a strong inflammation and destruction of the lung tissue for transmission and evolutionary survival [6,16]. This feature is unusual in most pathogens where virulence is not associated with their spread to other hosts [20]. “ should be modified. 

The phrases suggest that the tissue modification induced by MTBC are characteristic for MTBC, but the cell mediated response in described in case of Mycobacterium bovis and Mycobacterium avium. The lung is not the only organ that presents the modifications.

R: We believe that the referee misinterpreted what is written: the only species that is transmissible between humans is Mtb sensu stricto (the species mentioned in that sentence and not all MTBC) and must provoke lung destruction for successful transmission. Neither M. bovis nor M. avium are transmissible between humans. Extrapulmonary TB, including sensu stricto Mtb, is a dead end for the pathogen as it will not be transmissible.

  1. What is the connection between virulence factors and transmission to other hosts?

MTBC presence and consequent tissue modifications are described in several other species and the MTBC is known as a reverse zoonosis for both domestic and wild animals.

  1. In this review manuscript we refer only to human TB, therefore considerations regarding animal TB are outside the scope of this manuscript. In human TB, caused by reverse zoonosis of M. bovis, the morbidity to humans is much less aggressive and, in most cases, neither evolves into disease (only 2% of the world population has been affected). The main consequence of a compromised secretion level of ESAT-6 is the prevention of morbidity extension and transmission of M.bovis between humans.

  1. No link between the lines 43-55 and the aim of the study. 

Re: the capacity to generate a long live latency while maintaining the host healthy and the capacity to induce disease when transmission is required are the typical virulence factor meaning attributed to Mtb sensu stricto. For this successful ESAT-6 intervenes in all the phases mentioned between lines 43-55 and the link to the aim of the study.

  1. The authors do not underline the novelty of the study.

Re. the main novelty is to show in a manuscript this evaluation of ESAT-6 in the different phases of infection, including during the progression for active disease, necrosis and transmission, the latter being very recent findings after 2016. The reference to implications for therapeutic approaches is not the aim of the manuscript, but a consequence of the knowledge generated from ESAT-6: as stated the caution of using ESAT-6 or avoiding ESAT-6 in the design of new vaccines (attenuated) or subunit-vaccines, respectively, and in the design of new diagnostic tools for effective typification of the immune status of the patient.

  1. Lines 71-100 present a mixture of information

The authors should refer to ESAT-6 as a virulence factor in MTBC and present the differences found when comparing the genome of other species considered as TB etiological agents.

Re: We present ESAT-6 as a virulence factor of Mtb sensu stricto and the effects of RD8 and RD1 deletions on the ability of M.africanum and M.bovis to produce ESAT-6 in reduced extents. The direct consequence is a strong loss of fitness in humans reproduced in attenuation of disease and lost or low transmission between humans.

Regarding lines 71-100 they present a mixture of information needed and all related to the evolution of virulence among MTBC species with emphasis on the RD1 and RD8 and their impact on ESAT-6.

  1. 8.      Lines 124-350 should follow the introduction, while lines 71-100 should continue these

Re:

Line 124” ESAT-6 was first identified in 1995 by the discovery of a potent T-cell antigen in the short-term culture filtrate of Mtb [53,54]. The initial focus was on its potential use as a target for a TB vaccine to replace BCG [55]. Moreover, since patients and animals infected with MTBC respond strongly to ESAT-6 antigens, parallel emphasis was given to the development of a better diagnostic tool for TB [56].”

Lines 350 “Research on ESAT-6 and its involvement in several steps of Mtb pathogenesis, together with the strong antigenic recognition in TB patients, reveals its potential for therapeutic and diagnostic applications.”

The part of diagnostic and therapeutic interventions is not the  goal of the manuscript. It is presented in the irrespective subtitles as a consequence of ESAT-6 as a virulence determinant.

Lines 71-100 are practically all subtitle 2 and should be left as they are and not mix with information that is the main goal of the manuscript.

  1. lines 352-385 should be developed

RE: it corresponds to the whole part “4. ESAT-6 from a virulence factor to diagnostic tools and vaccines for TB”. As stated, the manuscript is not about comprehensive vaccine or diagnosis but only a consequence of deciphering the different levels of ESAT-6 as a virulence determinant. To avoid misunderstanding by future readers of the manuscript, we remove from the title “and a target for future interventions” . Likewise, it is described the avoidance of ESAT-6 in attenuated live vaccines (the only one based in Mtb just entered phase III of clinical trials) and its inclusion in subunit vaccines (which are now already in clinical phase trials). To clarify the requirements for diagnostic tools we rephrase the last sentences to: “It was originally designed to guide the preventive treatment of infected individuals at risk of developing active TB. Additionally, the ESAT-6-based IGRA allows differentiation between BCG vaccinated and unvaccinated individuals, as BCG does not possess or secrete these proteins. However, the challenges associated with ESAT-6-based subunit vaccines is that they will virtually override all modern available immunodiagnostics (IGRAs and skin tests), thereby limitating the ability to distinguish immunized from infected individuals. To overcome this ESAT-6 free IGRA assays are under development aimed at differentiating ESAT-6 subunit responses in vaccinated individuals.These assays are based on the combination of a cocktail of proteins that are part of the ESX-1 operon including the CFP-10 chaperon but excluding ESAT-6. Preliminary results show promising effects on the induction of IFNγ and the chemokine biomarker IP-10, allowing to distinguish subunit vaccinated from non-vaccinated individuals [56].  Future directions in diagnostic tools should allow to distinguish between latent TB infection, BCG/subunit vaccination, and active TB infection. This could revolutionize TB diagnostics and treatment strategies, allowing the development of differentiating biomarkers so relevant for assessing the status of immune activation and/or the stage of the infection.

Reviewer 2 Report

The present review is focused on the impact of the virulence factor ESAT6 on the success of the Mycobacterium tuberculosis complex (MTBC) pathogen, the leading or major cause of ill-health and mortality globally. Anes et al. include all aspect of ESAT6 as an unique mycobacterial virulence factor starting from the genetic background of the MTBC, followed by the biological consequences of these genetic variance  including the impact on vaccine development. This review is well written and appeals to a wide readership.

With regard to the genetic diversity within EsxA and EsxB, one small question is not answered sufficiently. What is the impact on single nucleotide polymorphisms within the RD1 or RD8 region ? Is there anything known with regard to different MTBC lineages?

Thank you!

Author Response

The present review is focused on the impact of the virulence factor ESAT6 on the success of the Mycobacterium tuberculosis complex (MTBC) pathogen, the leading or major cause of ill-health and mortality globally. Anes et al. include all aspect of ESAT6 as an unique mycobacterial virulence factor starting from the genetic background of the MTBC, followed by the biological consequences of these genetic variance  including the impact on vaccine development. This review is well written and appeals to a wide readership.

With regard to the genetic diversity within EsxA and EsxB, one small question is not answered sufficiently. What is the impact on single nucleotide polymorphisms within the RD1 or RD8 region ? Is there anything known with regard to different MTBC lineages?

Re: we thank the reviewer observations. It is beyond the scope of this manuscript to go into molecular microbiological details; the most important mutations are shown in Figure 1A and include single mutations on PhoPR and PhoP. Regarding BCG, reference 43 shows distinct insertional mutations (IS1660) on RD1 that affect the efficacy of different vaccinal strains of BCG in laboratories around the world. When referring to RD1 and RD8 in the present manuscript they are deletions on the genome comparative to Mtb and not point mutations (ref 43).

Reviewer 3 Report

This is fine and interesting review about biological properties ESAT-6. But from the title I expected to see large separate important chapter about molecules which inhibit Mtb virulence factors and I was very surprising did not see this part in the review. Therefore I suggest to rename title of this review, for example "ESAT-6 a major virulence factor of Mycobacterium tuberculosis" and corresponding rewrite abstract. 

Author Response

This is fine and interesting review about biological properties ESAT-6. But from the title I expected to see large separate important chapter about molecules which inhibit Mtb virulence factors and I was very surprising did not see this part in the review. Therefore I suggest to rename title of this review, for example "ESAT-6 a major virulence factor of Mycobacterium tuberculosis" and corresponding rewrite abstract. 

Re: we rename the title as suggested, but we think that the abstract as it is addressed the aim of the manuscript in emphasizing the role of ESAT-6 in the different phases of infection with Mtb sensu stricto. We did not find any review that puts together the relevance of ESAT-6 as virulence factor of Mtb contributing for its successful as human pathogen and the contextualization of the other MTBC that with compromised ESAT-6 production are less fitness in humans and not transmissible or practically not transmissible between humans.

The most recent publications after 2016 include ESAT-6 enrollment in the progression for active disease, necrosis and transmission. The reference to implications on therapeutic approaches is not the aim of the manuscript but a consequence of the knowledge generated from ESAT-6: as stated, the caution of using ESAT-6 or avoiding ESAT-6 in the design of new vaccines (attenuated) or subunit vaccines, respectively, and on the design of new diagnostic tools for effective typification of the immune status of the patient.

Reviewer 4 Report

In the paper under review the authors dedicated to ESAT-6 as a M.tb. virulence factor and TB therapeutic target. However, for all its relevance, there is an ambivalent impression after reading the article. The novelty and purpose of writing this review remains unclear. The most material discussed in the review is not new. The number of links including 2018 is just over 12%.

The idea of ​​using ESAT-6 for TB immunodiagnostics and vaccination is not new. The pioneer study that investigated the ELISpot assay performance in TB patients was completed by Lalvani et al. 2001. Attempts to create an anti-TB vaccine with ESAT-6 have been made for a long time - but this is not reflected in the review. Both protective and therapeutic vaccines are being developed. Currently no TB vaccine is more effective than BCG. It would also be important to mention that the creation of the ESAT-6 vaccine will practically cross out all modern immunodiagnostics (IGRAs and skin tests).

Minor:

line 32 - the emergence of multidrug-resistant strains to 70 year-old antibiotics - there is a drug resistance to relatively new antibiotics such as delamanid and bedaquiline.

line 33 - diagnostic based, in vast part of the world, on a 120-year-old microscopy technique - In most of the world (China, Russia and India), the main method of diagnosing TB is X-ray, followed by liquid culture and immunodiagnostics – IGRA and skin tests (as the main screening method). Diagnosis of TB by microscopy is possible only with a very advanced process.

line 274 - latency phase - Please explain what you mean by this. This chapter of the review describes the development of the immune response.

line 381-384 - Determining the difference between latent TB infection, BCG vaccination, and active TB infection could revolutionize TB diagnostics and treatment strategies, allowing the development of differentiating biomarkers so relevant to evaluate the status of immune activation and/or the stage of the infection – What does it mean? IGRAs cannot distinguish between active TB disease and LTBI.

Author Response

In the paper under review the authors dedicated to ESAT-6 as a M.tb. virulence factor and TB therapeutic target. However, for all its relevance, there is an ambivalent impression after reading the article. The novelty and purpose of writing this review remains unclear. The most material discussed in the review is not new. The number of links including 2018 is just over 12%. 

The idea of ​​using ESAT-6 for TB immunodiagnostics and vaccination is not new. The pioneer study that investigated the ELISpot assay performance in TB patients was completed by Lalvani et al. 2001. Attempts to create an anti-TB vaccine with ESAT-6 have been made for a long time - but this is not reflected in the review. Both protective and therapeutic vaccines are being developed. Currently no TB vaccine is more effective than BCG. It would also be important to mention that the creation of the ESAT-6 vaccine will practically cross out all modern immunodiagnostics (IGRAs and skin tests).

Re: The word TB and successful human-adapted pathogen means the ability to live long without causing disease in the host plus the capacity to be transmitted to other humans. Likewise, ESAT-6 is a major virulence of Mtb because in humans it intervenes in all relevant phases of the infection. The novelty of this review is based on the highlightning of these different phases in a single manuscript with the most recent publications regarding angiogenesis or induction of necrosis and survival during the caseating and cavitation phase, published after 2016. I did not find a single review published in all these data together.

We do agree that there’s nothing new yet on vaccines and diagnostic tests, but the current vaccines and tests are taking into account the role of ESAT-6. Accordingly, we have stated in the introduction that this part is only mentioned as a secondary consequence. Likewise, we rename the title removing “as a target for future interventions” to avoid misunderstanding in future readers and to focus the main goal of the manuscript on the relevance of ESAT-6 as a major Mtb virulence factor.

In part 4. We have now added the sentence “However, the challenges associated with subunit ESAT-6 based subunit vaccines is that they will virtually override all modern available immunodiagnostics (IGRAs and skin tests) thereby limitating the ability to distinguish immunized from infected people”.  and we thank the reviewer for the good suggestion.

Minor: 

line 32 - the emergence of multidrug-resistant strains to 70 year-old antibiotics - there is a drug resistance to relatively new antibiotics such as delamanid and bedaquiline.

Re: Delamanid and bedaquiline are still in phase 3 clinical trials and have just been approved by the FDA for the treatment of MDR-TB with no available alternatives. They are not yet approved as official antibiotics to treat TB. And they are restricted to this limited population plus those enrolled in clinical trials; so, comparing resistance to 70 years old antibiotics, resistance to delamanid and bedaquiline is irrelevant. And it is expectable that any antibiotic as soon available for therapy as soon resistance will be associated.

line 33 - diagnostic based, in vast part of the world, on a 120-year-old microscopy technique - In most of the world (China, Russia and India), the main method of diagnosing TB is X-ray, followed by liquid culture and immunodiagnostics – IGRA and skin tests (as the main screening method). Diagnosis of TB by microscopy is possible only with a very advanced process.

Re: We agree with the referee: as stated in the manuscript published in Lancet by Dedra 2016 (reference 4): “Despite suboptimal sensitivity (∼50%; detection limit of ∼104 organisms per mL), smear microscopy is the standard of care in most high-burden settings. However, we agree that the term “vast majority” is overstated. We now rephrase for: There exists also a need for better diagnostic tools to discriminate between different infection statuses status that can reach all geographical areas. In high burden geographic areas such as some regions of Africa, diagnosis is based on a 120-year-old, low-sensitive microscopy technique [4,5].

Round 2

Reviewer 1 Report

The authors issued detailed comments in regard to reviewer observations and recommendations, while the "revised" manuscript level is not sufficient  In fact, the "revised" manuscript remains a Copy paste material from papers that authors do not understand as they do not own sufficient knowledge to present and integrate basic words and concepts specific to fields such as epidemiology, etiology. microbiology ! The authors refer to etiology and epidemiology, zoonotic risk, but to reply it is not the aim of the study to present etiological agents of tuberculosis Please check WHO and WOAH websites to obtain a minimum level before you comment on Tuberculosis etiology, epidemiology and zoonotic risk This aspect is again for a basic level when writing a review paper - "The authors do not mention the review type, nor the criteria used for the selection of the presented information, please modify accordingly." The title modification and the detailed comments do not increase the level of the revised manuscript, the authors do not have a clear idea of their statements 

The English language for the lines marked with yellow needs editing

Author Response

Reviewer 1

The authors issued detailed comments in regard to reviewer observations and recommendations, while the "revised" manuscript level is not sufficient. In fact, the "revised" manuscript remains a Copy paste material from papers that authors do not understand as they do not own sufficient knowledge to present and integrate basic words and concepts specific to fields such as epidemiology, etiology. microbiology ! The authors refer to etiology and epidemiology, zoonotic risk, but to reply it is not the aim of the study to present etiological agents of tuberculosis Please check WHO and WOAH websites to obtain a minimum level before you comment on Tuberculosis etiology, epidemiology and zoonotic risk This aspect is again for a basic level when writing a review paper - "The authors do not mention the review type, nor the criteria used for the selection of the presented information please modify accordingly." The title modification and the detailed comments do not increase the level of the revised manuscript, the authors do not have a clear idea of their statements.

R: The reviewer points out that the authors produced “detailed comments in regard to reviewer observations and recommendations.” As such, the authors were expecting objective discussion and criticism. However, as this was not the case, and little objective criticism was provided beyond general statements and an attempt to phrase a provocation, we can only answer that we disagree with the general observations of the reviewer.

 Concerning the type of review, this manuscript is a narrative review (now stated in the abstract) and not a systematic review. The relevant bibliography was retrieved from Pubmed using “ESAT-6”, “ESAT-6 as virulence factor” and “Mtb pathogenesis”, “TB two-component system PhoPR”, from the last 15 years and the last 6 years. As is customary for this type of review, it also reflects the many decades of expertise of the authors in this area and the intimate knowledge of the work and researchers that produced it.

I was invited to participate in a special issue on Virulence factors in Mycobacterium tuberculosis infection: structural and functional studies. The journal Biomolecules refers to authors submit in the case of reviews, topics under their expertise.I emphasise in a review the most relevant interventions of ESAT-6 during the infection of Mycobacterium tuberculosis, and the amazing capacity to manipulate macrophages to be maintained in the host for decades driving the necrotic processes leading to transmission. As a “toxic” protein, the pathogen needs a strikingly control of their expression in the distinct phases of infection. Most of reviews are focusing on the secretion system apparatus and on the protein structure and combination of proteins required for secretion or instead on the role as immunogenic for diagnostic or sub vaccines unit and not under this perspective.

Consequently, 80% of the manuscript is devoted to ESAT-6 as a virulence factor in Mtb, 10% is now the application of the knowledge on ESAT-6 in vaccines and diagnostic tools (we extend the information relatively to the last version) and, 10% is devoted to introduce the reader to the discovery of ESAT-6 among MTBC with the molecular epidemiology studies that led to the characterization of the RD1 and RD8. The exchange of these regions among MTBC species allowed to demonstrate molecularly the Koch’s postulates necessary to prove the virulence associated to ESAT-6.

ESAT-6 can be evaluated as a virulence factor by the establishment in intracellular niches in macrophages, the intracellular survival along the infection, the ability to faster generate dead in animal models of infection, among others. The molecular exchange of RD1/8 between Mtb and M.bovis or M. africanum allowed to demonstrate the enrolment of these regions in the virulence, and at the end of ESAT-6. So, it does not make sense to move this part of the manuscript to the introduction and it is relevant enough to be in a subtitle following the introduction and the moving will disperse the introductory logical flux part information. The protein is also expressed in M. bovis and M. africanum  the second most described species causing TB in humans, and the differences among then and Mtb are those that result is less fitness demonstrated by those experiments and are supported by epidemiological studies as the gain of fitness of one M. bovis expressing more ESAT-6 that resulted in a outbreak in Spain in 1992 among HIV infected people, and the Beijing strain so successful transmitted among  humans.

Concerning a previous commentary, the reviewer asked to introduce the MTBC species and we did so in the previous version, and we now rephrase it in the present version to make it clearer.

The reviewer is welcome to gracefully provide their expertise on “etiology and epidemiology, zoonotic risk” to suggest the authors improvements to the relevance of the referred microorganisms to public health and eventual constructive corrections on the correct use of specific terms. We do remind that the focus of the review, to which most of the text body is dedicated, is not to replicate the excellent work of the WHO or its websites. Although interesting platforms to disseminate knowledge and engage the public, we believe that institutional websites should be used with parsimony as bibliographic material in writing a sound scientific document. As some statements may induce to misinterpretations in the way they were presented or under appreciation of the real zoonotic risk we rephrased the previous statement:

From: “TB is mainly a pulmonary infectious disease that is highly transmitted by the respiratory route by the species Mycobacterium tuberculosis (Mtb) [6]. Other pathogens can also infect and cause TB in humans with lower levels of disease morbidity and rarely transmitted from person-to-person”.

To:

Mycobacterium tuberculosis (Mtb) is considered the mainly causative agent of human TB [9]. Mtb infection primarily affects the lungs, a condition that contributes to the high transmissibility by the respiratory route[9]. Although not transmitted from person to person, extrapulmonary TB is a second form of disease manifestation that particularly affects less immunocompetent individuals, namely children or people co-infected with HIV [10-12]. In addition to Mtb, other pathogens can also infect humans and cause TB with similar clinical symptoms [13-20]”.

Nevertheless, in an attempt to address what we infer might be some of the reviewer’s concerns, we now include two sentences on this topic:

Although each of these animal MTBC variants causes TB in its host species, they may trigger slight or no disease outside of their adapted host, especially in immunocompetent hosts [13,30,31]. However, conditions such as geographic prevalence of infected animals, close humans contacts, route of transmission (milk, infected meet or airborne droplets containing pathogens), and conditions such as HIV infection or other immunosuppressive conditions should be taken in account to assess the true threat to public health [32]. These were included in the section of the manuscript that we believe does not interrupt the flow of the theme presentation and makes a good connection with the historical part.

The modifications on the present versions are highlighted in the manuscript. In the last part as mentioned we extended the information on vaccines and diagnosis. A phrase addressing the problematic of zoonotic and bovine TB was introduced as a limitation of current methods and the need for future investigations to solve this limitation.

Reviewer 4 Report

Unfortunately, I do not feel that the rebuttal from the authors fully addresses my original concerns. It would be better have read the materials of the WHO website before writing a review.

Author Response

Reviewer 4

Unfortunately, I do not feel that the rebuttal from the authors fully addresses my original concerns. It would be better have read the materials of the WHO website before writing a review.

Original concerns

The novelty and purpose of writing this review remains unclear. The most material discussed in the review is not new. The number of links including 2018 is just over 12%. 

The idea of ​​using ESAT-6 for TB immunodiagnostics and vaccination is not new. The pioneer study that investigated the ELISpot assay performance in TB patients was completed by Lalvani et al. 2001. Attempts to create an anti-TB vaccine with ESAT-6 have been made for a long time - but this is not reflected in the review. Both protective and therapeutic vaccines are being developed. Currently no TB vaccine is more effective than BCG. It would also be important to mention that the creation of the ESAT-6 vaccine will practically cross out all modern immunodiagnostics (IGRAs and skin tests).

R: The reviewer is concerned about the novelty of the manuscript since less than 12% of the references were published after 2018. We now add a few sentences with additional recent papers whose information reinforces the idea of the papers already included in the previous version. In addition, we expand the recent literature on the diagnostic and vaccine’s part.

Other concerns were related to Diagnosis and Vaccines. As we explained before, this is not the main objective of the manuscript but a consequence of the research on ESAT-6. The idea “of ​​using ESAT-6 for TB immunodiagnostics and vaccination is not new”. We agree that the concept is not new. However, the new therapeutic approaches and improved diagnostic tools under development still use ESAT-6 and, in this context, that is demonstrative of its relevant role in these innovations.

“Attempts to create an anti-TB vaccine with ESAT-6 have been made for a long time - but this is not reflected in the review. Both protective and therapeutic vaccines are being developed.”

We disagree that is not reflected in the manuscript: we mention (already in the first version) MTBVAC, the only one developed, that is designed not to secrete ESAT-6 but with all other Mtb immunogenic antigens stimulating the immune system and the subunit vaccines based on ESAT-6 together with the delivery systems already in clinical trials.

Because of their relevance, we now expand the state-of-the-art for vaccines by introducing the concept of therapeutic vaccines in addition to protective vaccines. In addition to the previously mentioned subunit vaccines designed to be both protective and therapeutic, we now include new ESAT-6-based subunit vaccines not yet in clinical phase trials. Indeed, we have expanded the information regarding diagnostic tools with the latest publications in the field including the problematic of diagnosis in children, HIV infected people and overall conditions were extrapulmonary TB could be a limitation as in bovine TB and the corresponding zoonotic infection. Modifications introduced in the present version are highlighted in the manuscript. We hope this addresses the concerns of the reviewer.

Regarding the last sentence: “It would be better have read the materials of the WHO website before writing a review”, its unobjective phrasing provides no context for us to give a better answer than acknowledging its attempt at a provocation.